# Methodological quality of cohort study on rheumatic diseases in China: A systematic review

**Huan Zhang[1,2]◉, Guoxiang Yi[1,2]◉, Mingzhu Dai[1,2], Yanping Li[2], Bin Wu◉[2]***

**1** Hunan University of Traditional Chinese Medicine, Changsha, China, **2** Department of Rheumatology, Chongqing Hospital of Traditional Chinese Medicine, Chongqing, China

◉ These authors contributed equally to this work.
* wuubinn@126.com

**Data Availability Statement:** All relevant data are within the paper and its Supporting Information files.

**Funding:** This study was financially supported by grants from the National Natural Science

## Abstract

### Objective

To evaluate systematically the quality of the cohort studies on rheumatic diseases in China.

### Methods

Relevant databases were searched to find cohort studies on rheumatic diseases in China, and the basic information included in the literature was extracted and analyzed. Chinese and English literature were then compared with regard to methodological quality, according to the Newcastle–Ottawa Scale (NOS).

### Results

In total, we included 46 cohort studies, with 19 studies published in English and 27 studies published in Chinese. With regard to the basic characteristics of the literature, 78.26% of the studies were published in the past four years; 16 studies were associated with hyperuricemia, followed by eight studies involving systemic lupus erythematosus. The sample size of the studies in Chinese was lower than that in English studies (*P*< 0.05). The English literature was superior to the Chinese literature in terms of informed consent, ethical review and selection of statistical analysis methods. The methodology quality of the 46 included studies showed that the English and Chinese NOS scores were 5.59 ± 1.25 and 6.06 ± 1.11, respectively, and the difference was significant *(P< 0.01)*. The "representativeness of the exposed group", "demonstration that outcome of interest was not present at start of study", and the "adequacy of follow up of cohorts" scores were relatively low in Chinese and English studies. The score for "was follow-up long enough for outcomes to occur" item in English was higher than that in the Chinese studies; however, the "study controls for the most important factor" score for Chinese papers was better than that for the English papers.

Foundation of China (no.:81673724), Chongqing Science and Technology Commission (cstc2018jxjl130084 and cstc2017jxjl130033), Chongqing municipal health and Health Committee (ZY20181009) and Chongqing Hospital of Traditional Chinese Medicine Advantageous Diseases Special Fund (cqszyyysbz2018004).The funders had no role in study design, data collection and analysis, decision to publish, or preparation of the manuscript.

**Competing interests:** The authors have declared that no competing interests exist.

## Conclusion

The Chinese rheumatic disease cohort studies started late, with a small sample size and fewer types of rheumatism. The quality of Chinese studies was better than English studies, and all reports were insufficient. In particular, "selecting exposed groups", "controlling the outcomes before study implementation" and "adequacy of follow-up" needed improvement.

## 1. Introduction

The cohort study is the second level of evidence in evidence-based medicine. It has more accessible data sources and lower costs than randomized controlled trials, and the results are more in line with clinical practice [1]. Its range of application has developed from health-related influencing factors to assessing the effectiveness of medical control measures [2]. Since the mid-19th century, many classic cohort studies have been carried out internationally. For example, a multicenter AIDS cohort study conducted in 1984 concluded that even if patients were seropositive for human immunodeficiency virus type 1 [HIV-1], those without AIDS would not develop *Pneumocystis carinii* pneumonia unless their CD4+ cells were depleted to 200 or fewer per cubic millimeter [3]. Another study, using data from 31,546 people in 40 countries, found that high-quality diets can reduce the incidence of cardiovascular events in people over 55 years of age [4]. Therefore, cohort studies play an important role in the assessment of risk factors, outcomes, and preventive measures for disease occurrence.

Rheumatism is an ancient disease encompassing nearly 200 diseases in 10 categories. the discipline was founded later but grew faster than many others [5]. In recent years, its increasing incidence and prevalence have attracted the attention of researchers all over the world. Cohort studies based on etiology and pathology, to guide clinical treatment, have become commonly used in the field of rheumatology [6–8]. For example, the Framingham cohort study investigating risk factors for osteoarthritis (OA) showed no significant relationship between inflammatory markers and OA [9,10]. In order to observe the risk of gout in people consuming fructose-rich beverages, 78,906 women were followed-up for 22 years, and the results showed that fructose-rich beverages increased the incidence of gout events [11]. China is a large country with many rheumatism patients, and some cohort studies have been carried out in recent years [12]. However, until now, their research status and quality have not been reviewed and evaluated.

This paper collects the literature on rheumatism cohort research conducted in China, discusses the progress of research in recent years, and evaluates the quality of research in the Chinese and English literature. The purpose of this study is to provide a reference for the improvement of future cohort studies.

## 2. Methods

### 2.1 Search strategy

The following databases were searched from their inception to January 2019. The English databases included PubMed, Web of Science, EMbase, and the Cochrane Library. The Chinese databases included the Chinese National Knowledge Infrastructure (CNKI), Wan Fang Database, the Chongqing Vip Information Database (VIP) and the Chinese Biomedical Database (SinoMed). For the English databases, the keyword searches used were "cohort studies", "rheumatic diseases", "autoimmune diseases" and "connective tissue diseases". The Medical Subject vocabulary (MeSH) searches used were "rheumatoid arthritis", "ankylosing spondylitis",

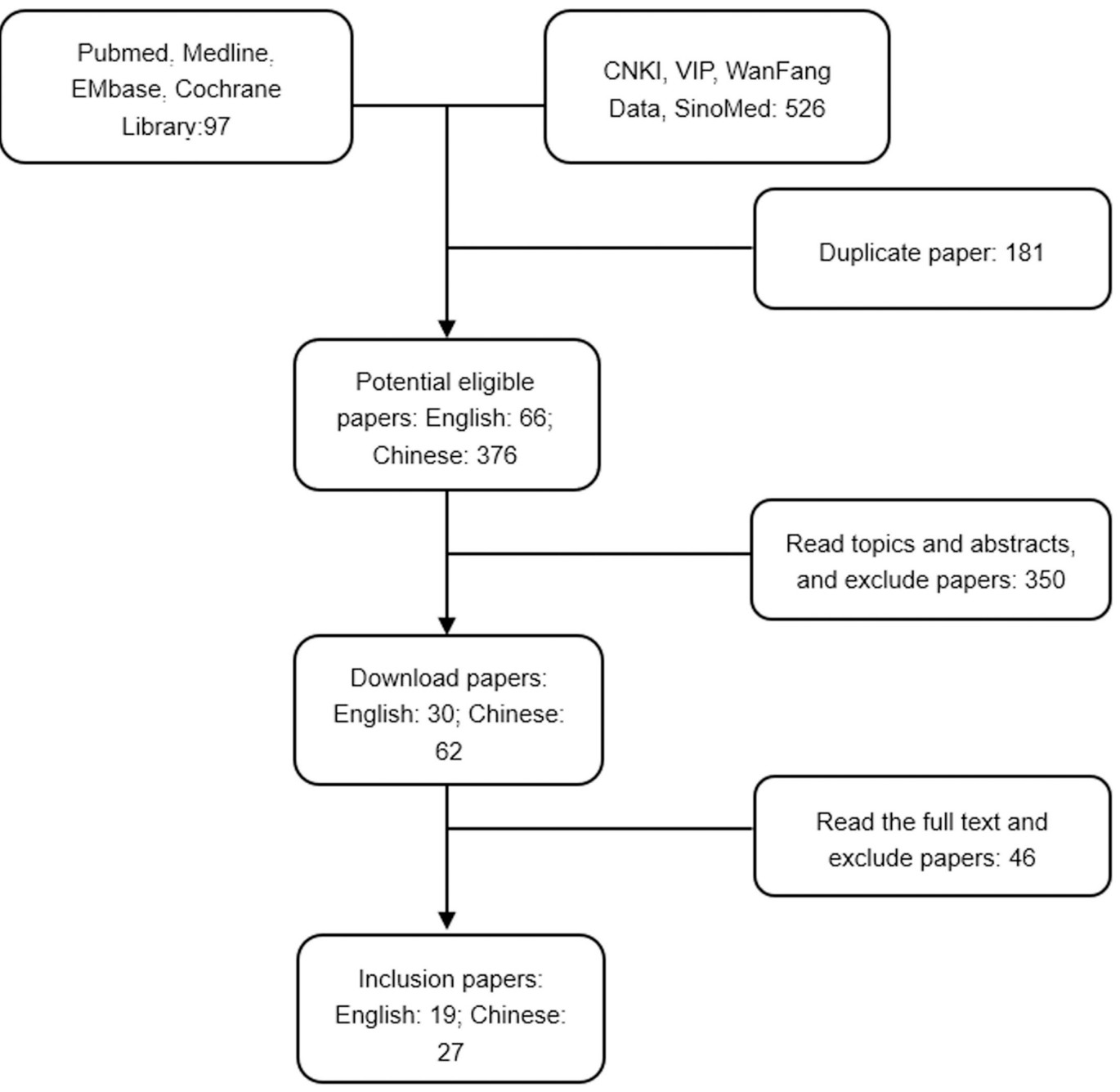

**Fig 1. Search and selection flow diagram.** A total of 97 English articles and 526 Chinese articles were retrieved. After screening and exclusion, 19 English and 27 Chinese papers were included.

"gout" and other common rheumatic diseases. For the Chinese-language searches, the same search strategy and search terms were used. The retrieval process is shown in Fig 1.

## 2.2 Study selection

The documents retrieved were imported into the NoteExpress document management software to exclude duplicate literature. By reading the topic and abstracts of the literature, two

researchers conducted a preliminary screening, and they subsequently read full text of each paper. Two researchers independently judged whether the inclusion criteria were met, and the divergences were referred to the third investigator for assistance.

## 2.3 Inclusion criteria

Studies that met the following criteria were selected for further analysis: the study type was a cohort study; the study participants were patients with rheumatic diseases from China; the study provided complete research data; languages were limited to Chinese or English; and the study was published in an official journal.

## 2.4 Exclusion criteria

Studies that met the following criteria were excluded: repeated published literature; reviews; systematic reviews; clinical control studies; randomized controlled trials; animal experiments; the full text remained unavailable after contacting the author.

## 2.5 Data extraction

Two reviewers (H Zhang and GX Yi) screened all titles and abstracts of the studies independently. Full texts of potentially-included studies were retrieved for further identification, according to the above criteria. A data extraction form was created, including the author, the year of publication, journal name, case collection location, sample size, disease researched, etc (Table 1). The disease was classified according to the *International Classification of Diseases (10th Edition)*. Disagreements were resolved by consultation with other authors, and a final decision was made through discussions and consultations.

## 2.6 Assessment of method quality

The study quality was evaluated according to the Newcastle–Ottawa Scale (NOS). The NOS is divided into selection (4 points), comparability (2 points) and outcome (3 points). There were eight items in three columns, and the final score was 0–9 points. The higher the score, the higher the quality. 0–4 points indicated low quality, and 5–9 points indicated high quality. Supplementary items for evaluation of methodological quality were as follows: design type of the study, calculation of sample size, informed consent and ethical review, statistical analysis methods, etc.

## 2.7 Statistical analysis

Statistical analysis was performed using SPSS version 23.0 (IBM Corp., Armonk, NY, USA). Data were presented as mean and standard deviation or frequency and percentage. Comparisons were conducted between the two groups. For normally distributed variables, means were compared using the *t*-test and nonparametric variables were analyzed using Pearson's chi-squared test. Two-tailed *P* values were used, with a $P < 0.05$ considered statistically significant.

# 3. Results

## 3.1 Search results

A total of 573 studies were retrieved. According to the aforementioned screening criteria and double assessment by two reviewers, we excluded 151 duplicate documents by reading topics and abstracts, removed 330 documents that did not meet the inclusion criteria, ruled out 46 non-conforming documents by reading the full text, and finally incorporated 46 articles,

**Table 1. List of basic characteristics of literature.**

| Study | Year of publication | Disease | Journal | Case collection location | Sample size | Funding | Informed consent | Ethical review |
|---|---|---|---|---|---|---|---|---|
| Adab P et al [13] | 2014 | RA | Rheumatology (Oxford) | Guangzhou | 7349 | | | N/A[a] |
| Liu Q et al [14] | 2015 | OA | Osteoarthritis Cartilage | Wuchuan | 1026 | N/A | N/A | N/A |
| Ma L et al [15] | 2018 | Gout | Clinical rheumatology | Qingdao | 5693 | N/A | | |
| Wang Y et al [16] | 2013 | Gout | Rheumatology international | Shandong Province | 659 | N/A | | |
| Wu J et al [17] | 2018 | SLE | BMJ open | Shanghai | 1352 | N/A | | |
| Wang Z et al [18] | 2018 | SLE | Lupus | Beijing | 260 | N/A | | |
| Yuan S et al [19] | 2013 | SLE | The Journal of rheumatology | Guangzhou | 3823 | | N/A | N/A |
| Zou YF et al [20] | 2013 | SLE | Inflammation | Hefei City | 260 | | | |
| Lin H et al [21] | 2012 | SLE | Clinical rheumatology | Chengdu | 158 | N/A | | |
| Xu W et al [22] | 2016 | Myositis | Immunology letters | Zaozhuang | 32380 | N/A | N/A | |
| Peng JM et al [23] | 2016 | Myositis | PloS one | Beijing | 109 | N/A | N/A | N/A |
| Chen D et al [24] | 2014 | Myositis | Clinical and experimental rheumatology | Guangzhou | 253 | N/A | | N/A |
| Tian Y et al [25] | 2015 | OP | Osteoporos Int | Wuhan | 38295 | | | |
| Liu H et al [26] | 2018 | KBD | Biological trace element research | Shanxi Province | 1214 | | | |
| Fan X et al [27] | 2017 | PBC | Scientific reports | Chengdu | 769 | N/A | | |
| Yang C et al [28] | 2017 | HUA | PLoS One | Luzhou | 4668 | N/A | | |
| Cui L et al [29] | 2017 | HUA | Mod Rheumatol | Tangshan | 101510 | N/A | | |
| Cao J et al [30] | 2017 | HUA | Int J Environ Res Public Health | Shandong Province | 58542 | | | |
| Villegas R et al [31] | 2010 | HUA | Metab Syndr Relat Disord | Shanghai | 3978 | | | |
| Wen J et al [32] | 2018 | OA | Chinese Journal of Clinical Healthcare | Anhui Province | 1904 | | N/A | N/A |
| Fang Y et al [33] | 2018 | RA | Chinese Journal of Clinical Healthcare | Hefei | 1812 | | | N/A |
| Zhao M et al [34] | 2016 | HUA | Hainan Medical Journal | Beijing | 126 | N/A | N/A | N/A |
| Zhang Y et al [35] | 2015 | AS | BMJ Chinese Edition | Shan Tou | 830 | N/A | | |
| Ji XJ et al [36] | 2016 | AS | Chinese Journal of Rheumatology | Beijing | 449 | N/A | N/A | N/A |
| Zhou W et al [37] | 2018 | HUA | Chinese Journal of Health Management | Changsha | 1859 | | | |
| Zhang CY et al [38] | 2014 | HUA | China Preventive Medicine | Beijing | 811 | N/A | N/A | N/A |

*(Continued)*

**Table 1.** (Continued)

| Study | Year of publication | Disease | Journal | Case collection location | Sample size | Funding | Informed consent | Ethical review |
|---|---|---|---|---|---|---|---|---|
| Jiang W et al [39] | 2018 | Myositis | Chinese Journal of Rheumatology | Beijing | 480 | | N/A | N/A |
| Wen J et al [40] | 2018 | AS | World Journal of Integrated Traditional and Western Medicine | Hefei | 399 | | N/A | N/A |
| Cao J et al [41] | 2017 | HUA | Journal of Shandong University (Health Science) | Shandong Province | 26006 | | N/A | N/A |
| Shou F et al [42] | 2010 | HUA | Prevention and Treatment of Cardio Cerebral Vascular Disease | Zhejiang Province | 374 | N/A | N/A | N/A |
| Li Y et al [43] | 2008 | HUA | CHINESE JOURNAL OF INTERNAL MEDICINE | Beijing | 1442 | | N/A | N/A |
| Gu X et al [44] | 2016 | OA | Chinese Journal of Joint Surgery (Electronic Version) | Shanghai | 362 | N/A | N/A | N/A |
| Zhang W et al [45] | 2002 | SLE | CHINESE JOURNAL OF RHEUMATOLOGY | Shanghai | 71 | N/A | N/A | N/A |
| Li D et al [46] | 2017 | OP | Chinese Journal of Osteoporosis | Zhongshan | 96 | | N/A | N/A |
| Li J et al [47] | 2016 | RA | CHINA HEALTH INSURANCE | Zhuzhou | 49 | N/A | | N/A |
| Mu X et al [48] | 2010 | RA | HEBEI MEDICAL JOURNAL | Beijing | 58 | N/A | N/A | N/A |
| Zhou Y et al [49] | 2016 | SLE | Journal of Applied Clinical Pediatrics | Shanghai | 60 | | | |
| Yang YF et al [50] | 2009 | SLE | MEDICAL JOURNAL OF NATIONAL DEFENDING FORCES IN NORTH CHINA | Taiyuan | 79 | N/A | N/A | N/A |
| Zhang C [51] | 2017 | HUA | World Journal of Complex Medicine | Zoucheng | 230 | N/A | N/A | N/A |
| Wu Y et al [52] | 2017 | HUA | Chinese Journal of Gerontology | Suzhou | 1628 | N/A | | N/A |
| Zha Z [53] | 2015 | HUA | Medical Information | Suzhou | 937 | N/A | N/A | N/A |
| Yang Y et al [54] | 2014 | HUA | Chinese Journal of Difficult and Complicated Cases | Wuhan | 266 | N/A | N/A | N/A |
| Lao Y [55] | 2015 | HUA | Chinese Primary Health Care | Suzhou | 994 | N/A | N/A | N/A |
| Liu X et al [56] | 2014 | HUA | Chinese Journal of Diabetes | Tangshan | 8603 | N/A | | N/A |
| Yuan YD et al [57] | 2017 | KD | Chinese Journal of Contemporary Pediatrics | Xuzhou | 404 | N/A | N/A | N/A |
| Ji XJ et al [58] | 2018 | AS | Chinese Journal of Internal Medicine. | Beijing | 897 | | | |

[a]N/A: Not Applicable. RA: Rheumatoid Arthritis; SLE: Systemic Lupus Erythematosus; OP: Osteoporosis; KBD: Kashin-Beck disease; PBC: Primary biliary cirrhosis; HUA: Hyperuricemia; OA: Osteoarthritis; AS: Ankylosing spondylitis; KD: Kawasaki disease.

including 19 English articles [13–31] and 27 Chinese articles [32–58]. The specific search process is illustrated in Fig 1.

## 3.2 Basic characteristics of the literature

**3.2.1 Disease classification.** A total of 11 diseases were included in the 46 articles, and hyperuricemia (16 articles) was the most studied, followed by systemic lupus erythematosus (8 articles). The specific disease distribution is shown in Fig 2.

**3.2.2 Year of publication.** The earliest research reports were published in 2002, and 36 studies have been reported in the last four years. The distribution of specific publication years is shown in Fig 3A.

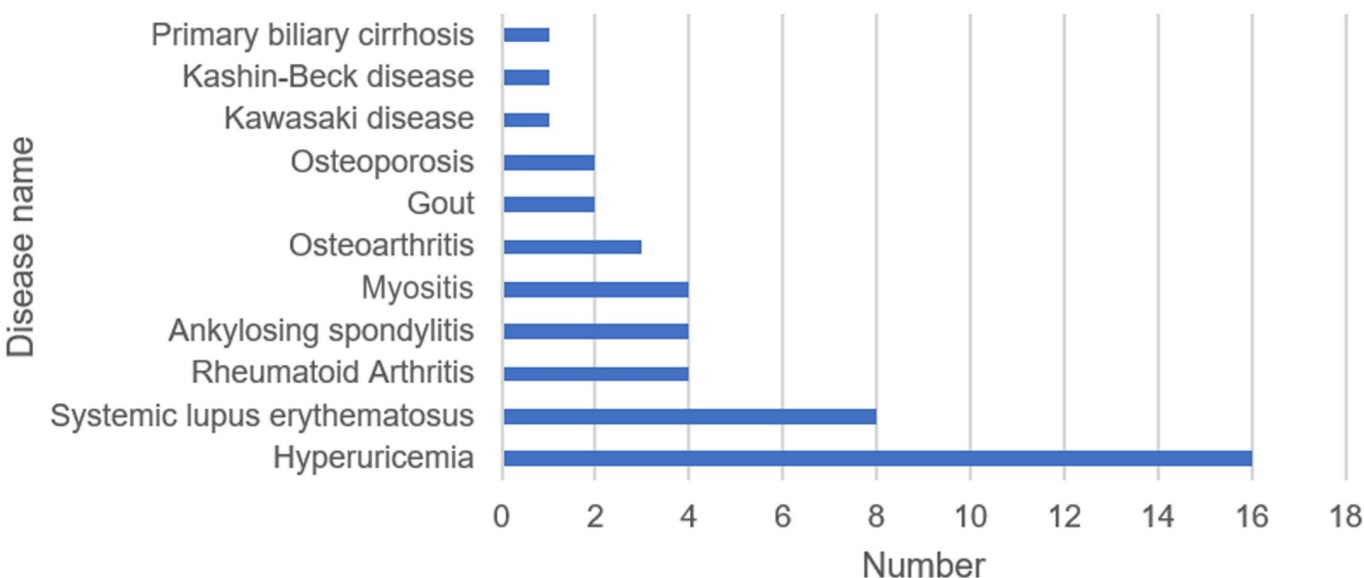

**Fig 2. Distribution of disease types.** Rheumatic disease cohort studies involve 11 disease types. The abscissa represents the quantity; the ordinate represents the disease type.

**3.2.3 Study types.** Sixteen prospective cohort studies were found [14,20,26,29,30,36,42, 45–47,51–56], which included 11 studies in Chinese and five studies in English. Moreover, 15 retrospective cohort studies were found [15,17–19,21–24,27,28,32,35,44,50,57], including 10 in English and five in Chinese. In addition, 14 papers did not describe the type of study and could not be judged [13,16,25,31,33,34,37–41,43,48,58], and there was only one ambispective cohort study [49]. The distribution of specific research types is shown in Table 2. It is worth noting that sample size calculation was mentioned in only one literature [40].

**3.2.4 Sample size.** A total of 313,564 subjects were included in the 46 studies, with a minimum of 49 cases and a maximum of 26,006 cases in Chinese, a minimum of 109 cases and a maximum of 101,510 cases in English. The sample size was less than 500 cases in 26.32% and 55.56% of English and Chinese studies, respectively. This difference was statistically significant (P< 0.05), and the specific sample size distribution is shown in Fig 3B.

**3.2.5 Informed consent and ethical review.** The informed consent and ethical review data are shown in Table 1. Informed consent reached 78.95% in English, but only 29.63% in Chinese. The difference between the two groups was obvious (P< 0.01). A total of 84.21% of English literature studies conducted ethical reviews, while only 14.81% conducted such reviews in the Chinese literature. As a result, statistically significant differences were found (P< 0.01).

**3.2.6 Statistical methods.** In this study, 73.68% of English studies and 55.56% of Chinese studies analyzed the baseline data, and only one reported data loss [35]. Three English and nine Chinese studies used only single-factor statistical analysis methods, such as single-factor variance analysis, t-tests, $\chi^2$ test, non-parametric testing, etc. One Chinese and one English study used Cox regression analyses to analyze effects over time; 15 English and 17 Chinese studies used logistic regression and Cox regression analysis to correct for the effects of confounding factors. The distribution of statistical analysis methods in Chinese and English studies is shown in Fig 4.

### 3.3 NOS methodological quality evaluation

The NOS [59] was used to evaluate the 46 articles included in the study. First, both Chinese and English studies had low scores for the "representativeness of the exposed cohort". Most of

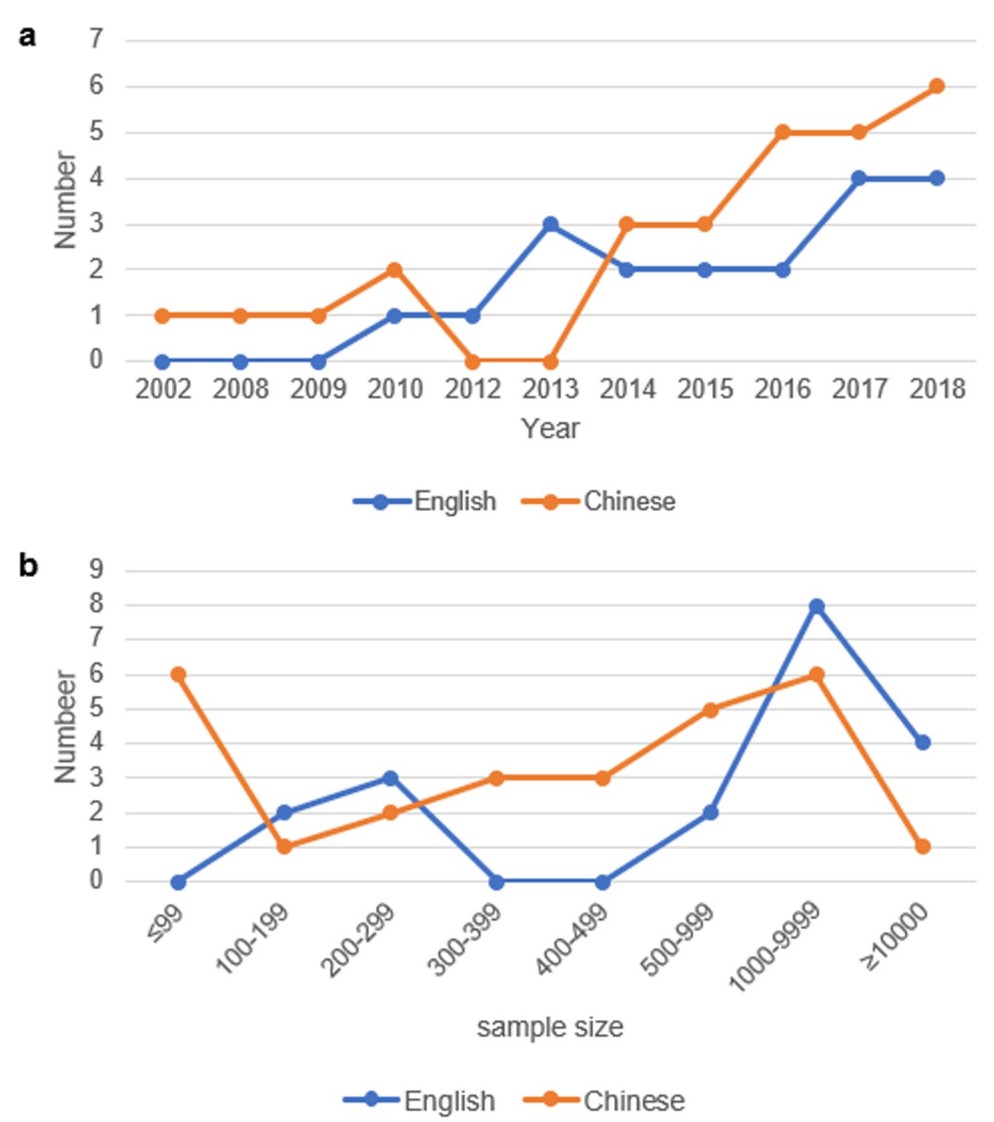

**Fig 3. Publication year and distribution of sample size. 3a.** Publication year of Cohort study literature on rheumatic diseases. The abscissa represents the year; the ordinate represents the quantity. **3b.** Distribution of sample size. The abscissa represents the number of samples; the ordinate represents the number of studies.

**Table 2. The distribution of literature types in the cohort studies.**

| Study Types | English | | Chinese | |
|---|---|---|---|---|
| | Number | Percent | Number | Percent |
| Prospective cohort study | 5 | 26.32 | 11 | 40.74 |
| Retrospective cohort study | 10 | 52.63 | 5 | 18.52 |
| Ambispective cohort study[a] | 0 | 0.00 | 1 | 3.70 |
| Could not be judged[b] | 4 | 21.05 | 10 | 37.04 |
| Total | 19 | 100.00 | 27 | 100.00 |

[a]Based on a retrospective cohort study, prospective observations continue for a period of time, which is a model that combines prospective cohort studies with retrospective cohort studies.

[b]The researches did not state the study type clearly and that could not be judged from the study.

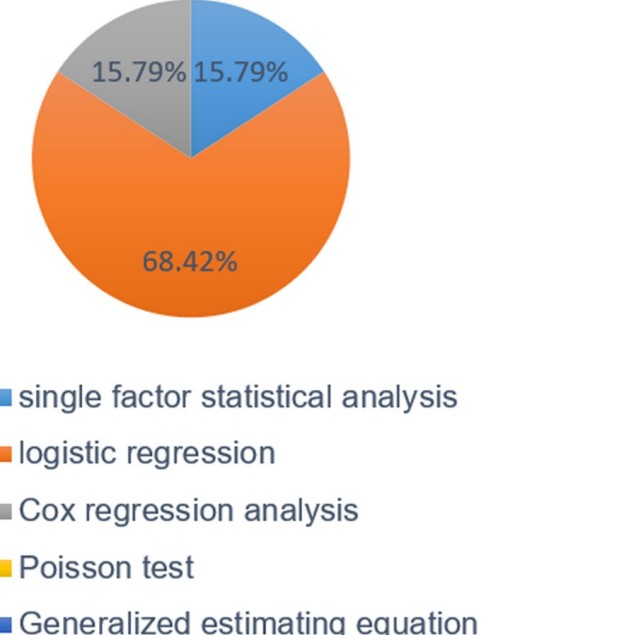
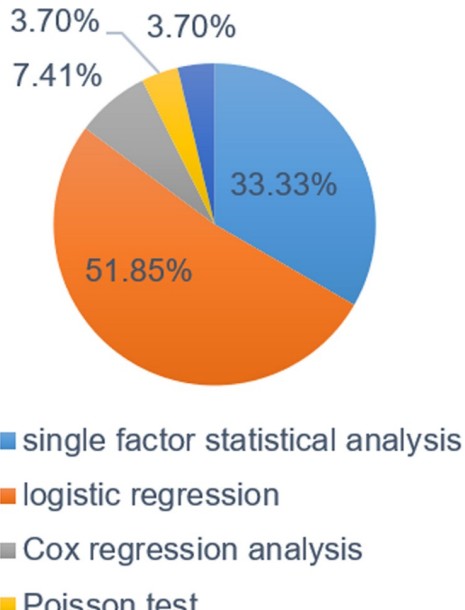

**Fig 4. Analysis methods. 4a.** Statistical analysis methods in English literature. **4b.** Statistical analysis methods in Chinese literature.

the studies were based on a specific group of people, such as a group of retired employees, medical examination staff of a company, and so on. Second, the English papers scored worse than the Chinese papers on the "comparability between groups" entries, because they did not control for important factors and confounding factors that could affect the results. Finally, the follow-up was described in only 13 English articles and eight Chinese articles. The Chinese literature scored lower than the English literature for follow-up time, and the follow-up time was generally insufficient. The scores for the "sufficient follow-up" entries in both groups were low. The specific evaluation scores are shown in Table 3.

According to the NOS evaluation criteria, the average score for the English literature was 5.59 ± 1.25, and the average score for the Chinese literature was 6.06 ± 1.11. The quality of the two groups was good, and the scores were mainly concentrated between 4–7. The average score of Chinese studies was higher than that of English studies. The difference between the groups was significant ($P< 0.01$). Table 4 gives the specific scores.

## 4. Discussion

An international cohort study on rheumatism is ongoing, and many large-sample multi-center clinical studies have been carried out. For example, the Framingham cohort study in the United States on the risk factors for osteoarthritis found that meniscal injury in the middle-aged and elderly populations increased with age and had no parallel relationship with clinical symptoms [60]. A retrospective cohort of 37,338 twins across Denmark showed that genes play only a minor role in the pathogenesis of rheumatoid arthritis [61]. Scholars at the University of St. Petersburg in Russia followed 498 women with systemic lupus erythematosus for 14 years. The research results suggested that early death was most common in those with active

**Table 3. Quality assessment of cohort studies using the Newcastle–Ottawa Scale.**

| Items | English | | Chinese | | $\chi^2$ | P |
|---|---|---|---|---|---|---|
| | Number | Percent | Number | Percent | | |
| **Selection** | | | | | | |
| 1) Representativeness of the exposed group | 6 | 31.58 | 7 | 25.93 | 0.176 | 0.675 |
| Selection of the non exposed cohort | 19 | 100 | 26 | 96.3 | 11.409 | 0.001[a] |
| Ascertainment of exposure | 16 | 84.21 | 23 | 85.19 | 0 | 1.000 |
| Demonstration that outcome of interest was not present at start of study | 7 | 36.84 | 12 | 44.44 | 0.266 | 0.606 |
| **Comparability** | | | | | | |
| Comparability of cohorts on the basis of the design or analysis | | | | | | |
| a) Study controls for the most important factor | 12 | 63.16 | 25 | 92.59 | 4.412 | 0.036[a] |
| b) Study controls for any additional factor | 12 | 63.16 | 21 | 77.78 | 1.176 | 0.278 |
| **Outcome** | | | | | | |
| 1) Assessment of outcome | 14 | 73.68 | 25 | 92.59 | 1.799 | 0.180 |
| Was follow-up long enough for outcomes to occur | 13 | 68.42 | 8 | 29.63 | 6.764 | 0.009[a] |
| Adequacy of follow up of cohorts | 7 | 36.84 | 9 | 33.33 | 0.061 | 0.805 |

[a]It represents statistical significance ($P < 0.05$)

lupus or concurrent infections, and late death was more common in patients with atherosclerotic coronary heart disease or acute myocardial infarction [62]. It is clear that cohort studies have played an important role in the prevention and treatment of rheumatism.

According to the analysis of the basic characteristics of the 46 studies evaluated herein, a total of 11 diseases were studied in all cohort studies. In the past four years, 78.26% the literature has been published, indicating that cohort research on rheumatism in China developed late, and the degree of emphasis has risen in recent years. Most reports described retrospective cohort studies, and 14 reports did not allow us to judge the type of research. Only one bidirectional cohort study was published, which may be related to its difficulty in implementation and high cost. The average sample size of studies in the Chinese literature was smaller than that of those in the English literature. Six Chinese studies had fewer than 100 participants. With small sample sizes, studies show shortcomings of under representation [63,64], leading to a low generalizability of Chinese cohort research. As can be learned from the statistical results, the large-scale Chinese reports with more than 500 cases accounted for 44.44%, while this percentage was 73.68% in the English reports. Of the 46 studies, only one reported an estimate of the sample size. With regard to informed consent and ethical review, the English literature was significantly better than the Chinese literature. However, 26.32% of English reports and 44.44% of Chinese reports did not describe baseline characteristics, which will affect the

**Table 4. The scores of cohort studies on the Newcastle–Ottawa items.**

| NOS | English | | Chinese | |
|---|---|---|---|---|
| Score | Number | Percent | Number | Percent |
| 3 | 1 | 5.26 | 0 | 0.00 |
| 4 | 2 | 10.53 | 6 | 22.22 |
| 5 | 7 | 36.84 | 5 | 18.52 |
| 6 | 4 | 21.05 | 6 | 22.22 |
| 7 | 4 | 21.05 | 9 | 33.33 |
| 8 | 1 | 5.26 | 1 | 3.70 |

interpretation of the results [65].Nine of the Chinese studies used only univariate analyses to compare differences between groups, and because there are no other factors controlled, this may result in bias in the results. In summary, the Chinese rheumatism cohort study developed late, involves fewer disease types, shorter follow-up time, and lack of multi-center large sample studies, which limit the theoretical and applied value of rheumatism cohort research in China. Therefore, it is particularly important to improve the quantity and quality of rheumatism cohort studies [66].

The NOS scale is a commonly used method for the evaluation of cohort studies [67]. In this paper, the 46 documents retrieved were evaluated using the NOS. First, the results show that the Chinese score was higher than the English score, although the overall scores were mainly between 4 and 7 points. This may have been related to the exclusion of non-standardized and low-quality papers during the process of selecting the literature, resulting in a higher overall score. Second, both the Chinese and English literature scored higher on items such as "selection of the non-exposed cohort", "ascertainment of exposure", and "assessment of outcome", but the score for "representation of the exposed group" was lower. The main reason for this is that the exposed group was generally a specific group, such as outpatients, inpatients or a group of employees, and cannot represent the entire population. In addition, both Chinese and English studies scored lower on "demonstration that outcome of interest was not present at start of study", because the paper did not explicitly explain or did not mention the relevant information. Third, 17 Chinese and nine English reports did not describe follow-up, and did not score on the item "whether the group was adequately followed". Finally, the Chinese studies were inferior to the English studies on "was follow-up long enough for outcomes to occur", because the Chinese studies did not specify or anticipate the required follow-up time and we were unable to determine whether the follow-up time was sufficient. However, Chinese studies scored better than English studies on the "select the most important factor" entry. The Chinese studies not only controlled important factors such as time, but also controlled other confounding factors such as age, gender, and duration of disease. Obviously, both the Chinese and English reports have different advantages and disadvantages, and it is necessary that researchers should learn from each other.

Although rheumatism cohort studies in China have shown a great deal of progress, there are still many areas that need improvement. For instance, researchers should try to avoid the choice of specific populations for exposure groups. Attention should be paid as to "Demonstration that outcome of interest was not present at start of study", so as not to affect the observation of the results. In addition, the follow-up time should be based on the time of occurrence of the observations, to estimate a sufficiently long follow-up time, and the follow-up process should be standardized and detailed. In addition, we should also pay attention to more rheumatic diseases that have not been studied, conduct multi-center large sample cohort studies as much as possible, improve informed consent and ethical review, and select appropriate statistical methods. The above methods can provide information for future rheumatism cohort studies and improve the standardization and quality of cohort research in China.

## 5. Conclusion

In this paper, we reviewed the progress and evaluated the quality of cohort studies on rheumatic diseases in China. On the one hand, the results showed that the Chinese rheumatic disease cohort studies developed late, with small sample sizes and fewer types of rheumatism. On the other hand, NOS quality assessment results show that the quality of Chinese reports is better than English reports. Some shortcomings were also discovered. In particular, "selecting exposed groups", "controlling the outcomes before study implementation" and "following up

groups" need to be improved. Therefore, well-designed multicenter and large-scale cohort studies are needed to improve clinical studies of rheumatic diseases in China.

## Supporting information

**S1 Checklist.**
(DOC)

**S1 Fig.**
(DOC)

## Author Contributions

**Conceptualization:** Huan Zhang, Bin Wu.

**Data curation:** Huan Zhang, Mingzhu Dai.

**Funding acquisition:** Bin Wu.

**Investigation:** Guoxiang Yi, Mingzhu Dai.

**Methodology:** Bin Wu.

**Project administration:** Yanping Li.

**Supervision:** Yanping Li, Bin Wu.

**Writing – original draft:** Huan Zhang, Guoxiang Yi.

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
