## [Decision Letter · Decision Letter 0]

6 Mar 2020

PONE-D-19-24922

Methodological Quality of cohort study on rheumatic diseases in China

PLOS ONE

Dear Prof Wu,

Thank you for submitting your manuscript to PLOS ONE. After careful consideration, we feel that it has merit but does not fully meet PLOS ONE’s publication criteria as it currently stands. Therefore, we invite you to submit a revised version of the manuscript that addresses the points raised during the review process.

The manuscript needs English editing by a native English speaker to improve the quality of language.

We would appreciate receiving your revised manuscript by Apr 19 2020 11:59PM. To enhance the reproducibility of your results, we recommend that if applicable you deposit your laboratory protocols in protocols.io, where a protocol can be assigned its own identifier (DOI) such that it can be cited independently in the future. For instructions see: http://journals.plos.org/plosone/s/submission-guidelines#loc-laboratory-protocols

We look forward to receiving your revised manuscript.

Kind regards,

Hua Zhou, Ph.D.

Academic Editor

PLOS ONE

Journal Requirements:

2. During our internal evaluation of your manuscript, we noticed several typos. For instances, line 37: Englishstudies, line 38: selectionofstatistical, line 176:Table 2.t. Please note that PLOS ONE does not copyedit accepted manuscripts, so please ensure the language in submitted articles is clear, correct, and unambiguous.

4. Please confirm that you have included all items recommended in the PRISMA checklist including identifying the study as a meta-analysis or systematic review in the title.

Additional Editor Comments (if provided):

The manuscript needs English editing by a native English speaker to improve the quality of language.

Reviewers' comments:

Reviewer's Responses to Questions

**Comments to the Author**

1. Is the manuscript technically sound, and do the data support the conclusions?

Reviewer #1: Yes

Reviewer #2: Yes

2. Has the statistical analysis been performed appropriately and rigorously? 

Reviewer #1: Yes

Reviewer #2: Yes

3. Have the authors made all data underlying the findings in their manuscript fully available?

Reviewer #1: Yes

Reviewer #2: Yes

4. Is the manuscript presented in an intelligible fashion and written in standard English?

Reviewer #1: Yes

Reviewer #2: Yes

5. Review Comments to the Author

Reviewer #1: Clinical research started late in China, and has been widely valued in recent years. However, there is still a certain gap with international standards in protocol design, data collection, data analysis and research report of the randomized controlled study or cohort study. In this paper, the author makes a systematic analysis of the cohort study of rheumatic diseases, and puts forward better suggestions for the future cohort study of rheumatic diseases, which basically achieve the research purpose of this paper.

In this paper, the author used standard evidence-based medicine research methods to retrieve, screen and extract the literature of rheumatic disease-related cohort studies in China, and evaluated the quality of the literature with NOS scale. On this basis, the author described the data profile of the cohort study, and compared the quality of literature through data analysis.

The data provided in this paper is sufficient, the statistical method is appropriate and the conclusion is reliable. According to the evaluation of English literature and Chinese literature respectively, the NOS scores show similar conclusions in " adequacy of follow up of cohorts " and are consistent with the clinical research practice in China.

In general, this paper reflects the current literature level of rheumatic disease-related cohort studies in China, and puts forward a number of reasonable suggestions, which basically achieve the purpose of the author's research.

Reviewer #2: This study collected the literature on rheumatism cohort researches conducted in China, and provided very important information about research quality comparision between Chinese and English publications. This is very good, especially in these days there are voices encouraging to publish paper in China Journal.

Some minor grammatical errors need to be revised before publication.

6. PLOS authors have the option to publish the peer review history of their article (what does this mean?). If published, this will include your full peer review and any attached files.

Reviewer #1: Yes: YaFeng Wang

Reviewer #2: Yes: Runyue Huang

---

## [Author Response · Author response to Decision Letter 0]

24 Mar 2020

Followings are the detailed responses for the comments. The blue words are comments and the black are reply. 

Major comments

http://www.journals.plos.org/plosone/s/file?id=wjVg/PLOSOne_formatting_sample_main_body.pdf and http://www.journals.plos.org/plosone/s/file?id=ba62/PLOSOne_formatting_sample_title_authors _affiliations.pdf

Response: We have browsed the website and checked our manuscript to ensure it meets PLOS ONE's style requirements.

2. During our internal evaluation of your manuscript, we noticed several typos. For instances, line 37: Englishstudies, line 38: selectionofstatistical, line 176:Table 2.t. Please note that PLOS ONE does not copyedit accepted manuscripts, so please ensure the language in submitted articles is clear, correct, and unambiguous.

Response: Sorry, this is our carelessness. 

Emendations: We have corrected them one by one.

line 27: the statement of “andanalyzed” was corrected as “and analyzed”;

line 34: the statement of “Englishstudies” was corrected as “English studies”;

line 35: the statement of “selectionofstatistical” was corrected as “selection of statistical”;

line 60: the statement of “pneumoniaunless” was corrected as “pneumonia unless”;

line 162: the statement of “Table 2.t” was corrected as “Table 2. It”;

line 234: the statement of “underrepresentation” was corrected as “under representation”.

line 240: the statement of “literature.However” was corrected as “literature. However”.

line 245: the statement of “developedlate” was corrected as “developed late”.

line 253: the statement of “points.This” was corrected as “points. This”.

line 254: the statement of “papersduring” was corrected as “papers during”.

3. Please include captions for your Supporting Information files at the end of your manuscript, and update any in-text citations to match accordingly. Please see our Supporting Information guidelines for more information: http://journals.plos.org/plosone/s/supporting- information. 

Response: All data and information have been presented in the manuscript, without additional information to supplement.

4. Please confirm that you have included all items recommended in the PRISMA checklist including identifying the study as a meta-analysis or systematic review in the title.

Response: We confirmed that we have included all items recommended in the PRISMA checklist and add “a systematic review” in the title.

5. The manuscript needs English editing by a native English speaker to improve the quality of language.

Response: Ok! To improve English expression, we invited International Science Editing to polish and edit this paper.

Emendations: The corrections in the paper are as following:

We have rewritten this part according to the requirements of the magazine. The comment symbol for the author's name was changed and Current Address was added. In addition, the information format of corresponding author has been changed.

We have fixed the font format for headings at all levels.

We upload our figure files to the Preflight Analysis and Conversion Engine (PACE) digital diagnostic tool to ensure that figures meet PLOS requirements. Besides, the titles, legends, comment marks of the pictures and tables were changed as required.

All references are supplemented as much as possible with doi and PMID, and Article 59 references have been updated.

We corrected three sentences.

Line 46: the sentence “The quality of the research was generally similar, and all reports were insufficient, in both the English and Chinese literature.” was corrected as “The quality of Chinese studies was better than English studies, and all reports were insufficient.”

Line 213: the sentence “A retrospective cohort of 37,338 twins across Denmark showed that genes play only a minor role in the pathogenesis of rheumatoid arthritis” was corrected as “the Framingham cohort study in the United States on the risk factors for osteoarthritis found that meniscal injury in the middle-aged and elderly populations increased with age and had no parallel relationship with clinical symptoms.”

Line 291: the sentence “On the other hand, the results of the NOS quality evaluation suggest that the quality of the study is similar in Chinese and English reports.” was corrected as “On the other hand, NOS quality assessment results show that the quality of Chinese reports is better than English reports.”

Some changes that do not affect the meaning of the content are not listed, but they are marked in red in the article.

---

## [Decision Letter · Decision Letter 1]

7 Apr 2020

Methodological Quality of cohort study on rheumatic diseases in China:  A Systematic Review

PONE-D-19-24922R1

Dear Dr. Wu,

We are pleased to inform you that your manuscript has been judged scientifically suitable for publication and will be formally accepted for publication once it complies with all outstanding technical requirements.

With kind regards,

Hua Zhou, Ph.D.

Academic Editor

PLOS ONE

Additional Editor Comments (optional):

Reviewers' comments:

Reviewer's Responses to Questions

**Comments to the Author**

1. If the authors have adequately addressed your comments raised in a previous round of review and you feel that this manuscript is now acceptable for publication, you may indicate that here to bypass the “Comments to the Author” section, enter your conflict of interest statement in the “Confidential to Editor” section, and submit your "Accept" recommendation.

Reviewer #1: All comments have been addressed

Reviewer #2: All comments have been addressed

2. Is the manuscript technically sound, and do the data support the conclusions?

Reviewer #1: Yes

Reviewer #2: Yes

3. Has the statistical analysis been performed appropriately and rigorously? 

Reviewer #1: Yes

Reviewer #2: Yes

4. Have the authors made all data underlying the findings in their manuscript fully available?

Reviewer #1: Yes

Reviewer #2: Yes

5. Is the manuscript presented in an intelligible fashion and written in standard English?

Reviewer #1: Yes

Reviewer #2: Yes

6. Review Comments to the Author

Reviewer #1: Same as the last recommendation， I think this paper meets the standards for publishing in PLOS ONE.

In this paper, the researchers used standard evidence-based medicine research methods to retrieve, screen and extract the literature of rheumatic disease-related cohort studies in China, and evaluated the quality of the literature with NOS scale. On this basis, the data profile of queue research is formed by data description, and the quality of literature is compared by data analysis.

The data provided in this paper is sufficient, the statistical method is appropriate. Whether published in English or in Chinese, the relevant NOS scores show similar conclusions in " adequacy of follow up of cohorts " and are consistent with the clinical research practice in China.

Reviewer #2: This is a good literature study to compare the quality between Chinese and English papers about Rheumatic diseases cohort. As the reasons addressed above, I suggest to accept this research article.

7. PLOS authors have the option to publish the peer review history of their article (what does this mean?). If published, this will include your full peer review and any attached files.

Reviewer #1: No

Reviewer #2: Yes: Runyue Huang

---

## [Editor Report · Acceptance letter]

10 Apr 2020

PONE-D-19-24922R1 

Methodological Quality of cohort study on rheumatic diseases in China:  A Systematic Review 

Dear Dr. Wu:

I am pleased to inform you that your manuscript has been deemed suitable for publication in PLOS ONE. Congratulations! Your manuscript is now with our production department. 

With kind regards,

on behalf of

Dr. Hua Zhou 

Academic Editor

PLOS ONE